# Diabetic Foot Complications: A Retrospective Cohort Study

**DOI:** 10.3390/ijerph20010187

**Published:** 2022-12-23

**Authors:** Bogdan Stancu, Tamás Ilyés, Marius Farcas, Horațiu Flaviu Coman, Bogdan Augustin Chiș, Octavian Aurel Andercou

**Affiliations:** 12nd Department of General Surgery, “Iuliu Hațieganu” University of Medicine and Pharmacy, 400012 Cluj-Napoca, Romania; 2Department of Molecular Sciences, “Iuliu Hațieganu” University of Medicine and Pharmacy, 400012 Cluj-Napoca, Romania; 3Hematology Department, Institute of Oncology “Prof. Dr. Ion Chiricuță”, 400015 Cluj-Napoca, Romania; 4Department of Vascular Surgery, County Clinical Emergency Hospital, 400347 Cluj-Napoca, Romania; 52nd Department of Internal Medicine, “Iuliu Hațieganu” University of Medicine and Pharmacy, 400012 Cluj-Napoca, Romania

**Keywords:** diabetic foot, diabetic foot ulcers, disease in lower extremity, gangrene, prevention, surgical treatment, wound healing

## Abstract

Diabetes mellitus is a highly prevalent disease globally and contributes to significant morbidity and mortality. As a consequence of multiple pathophysiologic changes which are associated with diabetes, these patients frequently suffer from foot-related disorders: infections, ulcerations, and gangrene. Approximately half of all amputations occur in diabetic individuals, usually as a complication of diabetic foot ulcers. In this retrospective study, we analyzed and characterized a cohort of 69 patients and their diabetes-related foot complications. The main characteristics of our cohort were as follows: older age at diagnosis (mean age 66); higher incidence of diabetes in males; predominantly urban patient population. The most frequent complications of the lower extremity were ulcerations and gangrene. Moreover, in our study, 35% of patients required surgical reintervention, and 27% suffered from complications, while 13% required ICU admission. However, diabetic foot lesions are preventable via simple interventions which pointedly reduce foot amputations. Early identification and the appropriate medical and surgical treatment of the complications associated with diabetic foot disease are important because they still remain common, complex and costly.

## 1. Introduction

There are more than 463 million people with diabetes mellitus (DM) worldwide [1]. These patients are at risk of multiple diabetes-related complications. Specifically, foot problems cause serious morbidity and mortality in these patients. Diabetic patients frequently suffer from foot-related disorders such as gangrene, infection and ulcerations. DM is a significant risk factor for peripheral artery disease [2,3]. The International Diabetes Federation estimates that 9.1–26.1 million diabetic individuals will develop diabetic foot ulcers (DFUs) every year [4]. Moreover, type 1 and type 2 diabetics have a lifetime risk of foot ulcers as high as 34 percent [4]. Epidemiological data indicate that diabetic patients have a risk of 2.5% per year to develop foot ulcers [5]. As shown by a population-based cohort study, diabetic foot ulcers are associated with 5% mortality within the first 12 months and 42% percent mortality at 5 years after first occurrence [6]. Moreover, individuals living with diabetic foot ulcers have a lower quality of life and greater morbidity [7].

In the United States, the treatment of the diabetic foot is responsible for approximately 30 percent of the total costs for diabetic patients’ care—an estimated U.S. $ 176 billion in healthcare costs [8]. Additionally, about 20% of patients have unhealed diabetic foot lesions at one year after their diagnosis [9], and the recurrence rate is approximately 40% at one year [4]. Diabetic foot amputations also carry significant stigma and impact the quality of life of DM patients [10].

In Romania, many patients ask for medical care within one to six months after the onset of the symptoms of DM or its complications. The presence of diabetic foot complications such as ulcers and amputations has also been shown to increase the cost of hospitalization with more than 40% compared to patients without complications (€ 724 vs. € 517) [11].

Foot amputations in DM patients are usually progressive and often recurrent [12,13].

Moreover DFU can lead to gangrene, infection or amputation and are a significant cause of morbidity; they account for approximately two-thirds of all non-traumatic foot amputations in the United States [14].

There are different procedures reported in literature to avoid these complications, in particular plantar foot ulcers and Charcot foot as minimally invasive distal metatarsal diaphysis osteotomies and tibiotalocalcaneal arthrodesis using a retrograde intramedullary nail for chronic plantar DFU [15].

Multiple risk factors contribute to the development of DFUs and other foot lesions such as gangrene and lower extremity infections: poor glycemic control; vascular disease; inadequate foot care; neuropathy and the subsequent loss of protective sensation; foot deformity; trauma; diabetes-related compromised immunity and infections [16].

In this retrospective cohort study of 69 patients, we analyze multiple parameters of patients with diabetic foot lesions (complication rates, bacteriology, intensive care admission, etc.) and their respective outcomes. Our study aimed to evaluate the role of age, foot ulcer bacteriology, reintervention complications and reintervention rate and ICU admission in treatment of DFU patients in order to prevent amputations.

## 2. Materials and Methods

### 2.1. Data Acquisition

Patients with type 2 diabetes complications necessitating surgery admitted to the Number 2 Surgical Clinic of the County Clinical Emergency Hospital of Cluj-Napoca between March 2019 and February 2021 (3 years) were included in this study.

Information regarding age, gender, living environment, duration of hospital stays, diagnosis, pre-operative bacteriological assay lesion exudate, post-operative complications, reintervention or ICU admission were pulled from the hospital data base.

Records of patients of all ages with type 2 diabetes, admitted for either pre-scheduled or emergency surgery related to type 2 diabetes complications, were included in this study. Records from patients that were only admitted on an outpatient basis or who died during their hospital admission as a result of post-operative complications or complications related to type 2 diabetes were excluded.

### 2.2. Statistical Analysis

Statistical analysis was carried out using RStudio Desktop (RStudio© PBC v2021.09.0 Build 351, Posit PBC, Boston, MA, USA) while graphical representations of data were generated using Excel from Microsoft 365 for Windows (Microsoft^®^ Excel^®^ for Microsoft 365 MSO v2204 Build 16.0.15128.20158, Microsoft, Washington, WA, USA). A value of *p* < 0.05 at both tails was considered statistically significant.

Quantitative variables were presented as mean ± standard deviation for normally distributed data and median with minimum and maximum for non-normally distributed data. The normality of quantitative data was evaluated using the Shapiro-Wilk test. The correlations between quantitative variables were analyzed using the Mann-Whitney U test (Wilcoxon rank-sum test). For qualitative data, Odds Ratio (OR) and Relative Risk (RR) were calculated.

## 3. Results

### 3.1. Descriptive Statistics

Our study included a total of 69 patient records, the characteristics of which are presented in Table 1, Table 2 and Figure 1 below.

### 3.2. Study Results

Hospitalization duration was significantly higher in patients < 68 years old, in patients who developed post-operative complications, as well as in the case of patients who required reintervention (Table 3).

Bacteriological assay results did not differ significantly between patients with ages < 68 years compared to patients > 68 years old. Patients < 68 years old were 2.6 times more likely to require reintervention than patients > 68 years old (Table 4).

There were no statistically significant differences between men and women regarding post-operative complications, bacteriology assay results, reintervention, or ICU admission.

Patients living in an urban environment were 1.87 times more likely to have a positive bacteriological assay result compared to patients living in a rural environment (Table 5).

Patients who developed post-operative complications were 2.11 times more likely to require reinterventions than those that did not develop complications (Table 6).

There were no significant differences between patients with positive and negative bacteriological assay results regarding reintervention or ICU admission. There was no significant difference in ICU admission between patients who suffered reintervention compared to those that did not.

## 4. Discussion

The Society for Vascular Surgery has proposed a new classification system for the threatened lower limb, based on the three main factors that have an impact on limb amputation risk: Wound (W), Ischemia (I) and foot Infection (“fI”)—the WIfI classification. The system also covers diabetic patients, previously excluded from the concept of critical limb ischemia because of their complex clinical condition. The classification’s purpose is to provide accurate and early risk stratification for patients with threatened lower limbs: assisting with clinical management, enabling comparison of alternative therapies, and predicting risk of amputation at 1 year and the need for limb revascularization [17].

Foot ulcers and other lower limb lesions frequently affect diabetics during their lifetime. Age is a vital determinant of susceptibility to complications. Data from multiple studies show that the incidence of diabetic foot complications (diabetic foot lesions, infections and amputations) increase with diabetes duration and patient age [16,17]. The mean age of the patients in our study was 66. This data is in line with other studies which show that advanced age and duration of diabetes independently predict diabetes-related mortality and morbidity rates [18,19].

However, the incidence of diabetes is also increasing in the younger population. In our study, the duration of hospitalization was significantly higher in patients < 68 years old. This may be related to the fact that individuals who develop complications at a younger age might also have developed DM earlier in their lives or might suffer from a more severe, uncontrolled form of DM and thus sustained prolonged circulatory injuries from uncontrolled glycemia. Moreover, in our study, younger patients (<68 years old) had a 2.6 times higher risk of surgical reintervention than those over 68 years of age.

There is a disparity in the rate of incidence of diabetes-related complications in urban, suburban and rural areas. However, research connecting patient whereabouts and diabetes-related complications is rare. In our study, there was an urban majority of patients admitted to the surgical ward. Data from other studies indicates that DM incidence is rapidly increasing in the urban population of developing countries and in elderly individuals (>65 years old) [20]. A systematic analysis showed that the prevalence of type two DM was associated with higher age, urban residence and male gender [21]. Another study highlighted a higher prevalence of diabetes in individuals with a high socioeconomic status and urban livelihood compared to rural areas with a low socioeconomic status [22]. Thus, data from these studies indicate that aging populations and urbanization are strong drivers of the increasing prevalence of DM.

In our study, the majority of individuals with diabetes were males (74%). This is in line with data from other studies which show that diabetes incidence and prevalence and diabetes-related complications (gangrene, amputation) are higher in males [23,24,25]. In our study, 10% of patients presented Charcot arthropaty. Males have higher pressures at the level of the foot and a more limited joint flexibility (higher mean height and weight and more frequent peripheral neuropathy) [26,27]. Multiple reasons may account for the higher incidence of diabetes and its complications in males. Women take a proactive stance in caring for their bodies showcasing higher levels of self-care and have a positive body-care mood: they are more involved in preventative and self-care activities; they inform themselves more often than males and try to adapt themselves to the situation. In comparison, men have a passive attitude, a dependency tendency (help from relatives or spouses) and they expressed negative attitudes and fear more often than females [28]. Moreover, males are more frequently exposed to trauma and wear inadequate footwear [29,30]. This may partially account for the differences observed between genders.

Diabetic foot lesions are susceptible to gangrene, ulceration and other complications. A retrospective registry-based study indicated that diabetic foot complications ensued in 3.3% of the studied patients; out of that 2.05% were DFUs [25], which is similar to the international range [31,32]. Amputation frequency is estimated to occur in 1.06–3% of diabetics, depending on the studied cohort [33,34]. Once the lesion has occurred, the aim is to preserve viable tissue and prevent further ischemia and infections [35]. The most significant complication in our patient cohort was gangrene (68%), followed by ulcerations of the foot (22%). Gangrene is a debilitating complication in diabetic patients, and many of these patients require different level amputations. This is in line with other data sources which point to foot ulceration and wounds, amputations and infections as frequent complications in diabetic patients [36].

With suitable therapy (surgical debridement, preventing and treating infections, vascular reconstructions), foot lesions heal and the requirement for amputation and reintervention is averted [37,38]. Approximately 40% of patients have an ulcer recurrence at one year after the healing of the initial lesion, and 60% have an ulcer recurrence at three years. As such, wound closure in diabetic foot patients should be looked at as remission rather than healing [39]. In our study, 35% of patients required surgical reintervention and 27% suffered from complications, while 13% required ICU admission. Other studies show a lower unplanned 30-day ICU readmission rate [4]. This discrepancy may be related to the fact that, according to the State of Health in the EU, a report released by the European Commission, Romania has among the highest rates of death from treatable and preventable causes compared to other EU countries. In Romania, Mota et al. reported a prevalence of diabetes of 11.6%, based on the results of an epidemiological study conducted between 2012 and 2014; more than 1,200,000 Romanian adults in the 20–79 years age range are living with this disease [40]. Moreover, a substantial proportion of the population has unmet medical needs. As such, higher rates of complications and ICU admissions are to be expected.

Rastogi et al., in a multicenter 14 years prospective study of 2880 patients, evaluated limb amputation and mortality after the first neuropathic diabetic foot ulcer (DFU). They concluded that every one in three individuals with neuropathic DFU has amputation, and every sixth individual has an early demise. Prevalent nephropathy and incident amputation following DFU predicts mortality [41].

Sohn et al., compared the risks of lower-extremity amputation between patients with Charcot arthropathy and those with diabetic foot ulcers, and they found at the end of follow-up, that among patients < 65 years old, amputation risk relative to patients with Charcot alone was 7 times higher for patients with ulcer alone and 12 times higher for patients with Charcot and ulcer [42].

Five year mortality, reported by Armstrong et al., for Charcot, DFU, minor and major amputations were 29.0, 30.5, 46.2 and 56.6%, respectively. This is compared to 9.0% for breast cancer and 80.0% for lung cancer. Five-year pooled mortality for all reported cancer was 31.0% [43].

Among our patients, 39% presented a positive bacteriology assay, and the most common bacteria isolated was *S. aureus*. Atlaw et al. performed an institutional-based multicenter, cross-sectional study in selected Hospitals in Addis Ababa, Ethiopia, from November 2020 to May 2021. A sterile swab was used to collect samples from the foot ulcer and a sterile needle to collect pus. The majority of bacteria isolated from patients presenting with diabetic foot ulcer infections were found to be multi-drug resistant, and they demonstrated the importance of timely identification of infection of diabetic foot ulcers, proper sample collection for identification of the pathogens and for determining their antibiotic susceptibility pattern before initiating antimicrobial treatment [44].

However, foot ulcerations are preventable via simple interventions which significantly reduce foot amputations [45]. Identifying the key risk factors which lead to foot complications in diabetic patients and setting up prevention programs can significantly improve patients’ outcomes and quality of life and reduce economic burden for healthcare systems [46].

Multiple limitations need to be taken into account when considering this study: there is an absence of data on confounding factors (smoking, alcohol usage, exercise, and other comorbidities); retrospective studies provide inferior levels of evidence in comparison to prospective studies. The sample size is limited, and data analysis did not always reach statistical significance. We also did not evaluate the WIfI grade of our patients. We are looking forward to performing further prospective studies about DFU.

## 5. Conclusions

Early identification and the appropriate medical and surgical treatment of the complications associated with diabetic foot disease are important because they still remain common, complex and costly. Thirty-five percent of patients with diabetic foot complications (infections, ulcerations, gangrene) required surgical reintervention and 27% suffered from complications, while 13% required ICU admission. The following factors have been found relevant for diabetic foot complications: older age (mean age 66), males and urban patient population.

## Figures and Tables

**Figure 1 ijerph-20-00187-f001:**
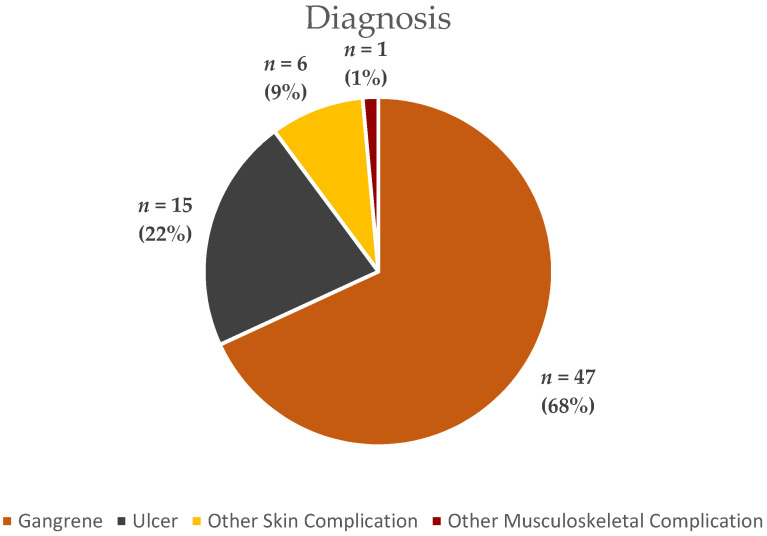
Types of complications from patient records.

**Table 1 ijerph-20-00187-t001:** Quantitative variables from patient records.

Variable	Number of Records	Mean ± SD Median (Min–Max)
Age (years)	*n* = 69	66.20 ± 1.33
Hospitalization (days)	*n* = 69	9 (1–49)

**Table 2 ijerph-20-00187-t002:** Qualitative variables from patient records regarding gender, home environment, lesion bacteriological assay results, official diagnosis category, complications, reintervention, and ICU admission.

Variable		Number of Records (Percent of Total)
Gender	Male	*n* = 51 (74%)
	Female	*n* = 18 (26%)
Environment	Urban	*n* = 34 (52%)
	Rural	*n* = 32 (48%)
Complications	Yes	*n* = 16 (27%)
	No	*n* = 44 (73%)
Bacteriology	Positive	*n* = 30 (46%)
	Negative	*n* = 35 (54%)
Reintervention	Yes	*n* = 24 (24%)
	No	*n* = 45 (65%)
ICU admission	Yes	*n* = 9 (13%)
	No	*n* = 60 (87%)

Abbreviations: ICU, intensive care unit.

**Table 3 ijerph-20-00187-t003:** Median hospitalization duration in days between different groups.

Group		Median Hospitalization Duration (Days)	*p*
Age (years)	<68	12	<0.01 *
	>68	8	
Post-operative complications	Yes	17.5	<0.01 *
	No	8.5	
Bacteriology	Positive	10.5	0.291
	Negative	9	
Reintervention	Yes	15	<0.01 *
	No	8	
ICU admission	Yes	11	0.555
	No	9	

Significant differences noted with *. Abbreviations: ICU, intensive care unit.

**Table 4 ijerph-20-00187-t004:** Comparison of post-operative complications, bacteriology assay results, reintervention, and ICU admission between patients aged < 68 years and > 68 years.

Parameter		Age (Years)		OR	RR	*p*
		<68	>68			
Post-operative complications	Yes	*n* =11	*n* = 5	2.895 (95%CI, 0.860–9.745)	2.2 (95%CI, 0.809–6.718)	0.143
	No	*n* = 19	*n* = 25			
Bacteriology	Positive	*n* = 15	*n* = 15	1.059 (95%CI, 0.399–2.808)	1.031 (95%CI, 0.571–1.855)	1
	Negative	*n* = 17	*n* = 18			
Reintervention	Yes	*n* = 17	*n* = 7	4.402 (95%CI, 1.508–12.847)	2.649 (95%CI, 1.212–6.286)	<0.01 *
	No	*n* = 16	*n* = 29			
ICU admission	Yes	*n* = 5	*n* = 4	1.429 (95%CI, 0.349–5.847)	1.364 (95%CI, 0.338–5.764)	0.728
	No	*n* = 28	*n* = 32			

Significant differences noted with *. Abbreviations: OR, odds ration; RR, relative risk; CI, confidence interval; ICU, intensive care unit.

**Table 5 ijerph-20-00187-t005:** Comparison of post-operative complications, bacteriology assay results, reintervention, and ICU admission between patients living in urban and rural environments.

Parameter		Environment		OR	RR	*p*
		Urban	Rural			
Post-operative complications	Yes	*n* =10	*n* = 5	2.421 (95%CI, 0.705–8.312)	1.931 (95%CI, 0.695–5.984)	0.23
	No	*n* = 19	*n* = 23			
Bacteriology	Positive	*n* = 20	*n* = 10	3.333 (95%CI, 1.174–9.462)	1.875 (95%CI, 1.022–3.558)	<0.05 *
	Negative	*n* = 12	*n* = 20			
Reintervention	Yes	*n* = 10	*n* = 13	1.642 (95%CI, 0.592–4.557)	1.381 (95%CI, 0.656–2.965)	0.44
	No	*n* = 24	*n* = 19			
ICU admission	Yes	*n* = 3	*n* = 5	1.914 (95%CI, 0.418–8.762)	1.771 (95%CI, 0.394–9.031)	0.469
	No	*n* = 31	*n* = 27			

Significant differences noted with *. Abbreviations: OR, odds ration; RR, relative risk; CI, confidence interval; ICU, intensive care unit.

**Table 6 ijerph-20-00187-t006:** Comparison of bacteriology assay results, reintervention, and ICU admission between patients that developed post-operative complications and those that did not.

Parameter		Post-Operative Complications		OR	RR	*p*
		Yes	No			
Bacteriology	Positive	*n* = 8	*n* = 19	1.444 (95%CI, 0.444–4.696)	1.207 (95%CI, 0.566–2.087)	0.564
	Negative	*n* = 7	*n* = 24			
Reintervention	Yes	*n* = 10	*n* = 13	3.974 (95%CI, 1.195–13.216)	2.115 (95%CI, 1.022–3.659)	<0.05 *
	No	*n* = 6	*n* = 31			
ICU admission	Yes	*n* = 4	*n* = 3	4.556 (95%CI, 0.893–23.235)	3.667 (95%CI, 0.742–19.449)	0.074
	No	*n* = 12	*n* = 41			

Significant differences noted with *. Abbreviations: OR, odds ration; RR, relative risk; CI, confidence interval; ICU, intensive care unit.

## Data Availability

Not applicable.

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
