# Peer review of "Diabetic Foot Complications: A Retrospective Cohort Study"

_ijerph, 2022, doi:10.3390/ijerph20010187_

Round 1
Reviewer 1 Report
Review
Many thanks to the authors for having presented a so interesting study about “Diabetic foot complications: a retrospective cohort study”.
Before resubmitting the revision version of the article, text is well aligned, change the text size of the background since it is smaller the other sections. Language is not good enough, need to be correct by an English mother tongue.
Abstract
The abstract is well structured, and it contains the main information of the study.
Key words
Please provide them in alphabetic order.
Background
The introduction identifies the problem that is being addressed in the manuscript, but I cannot clearly understand the purpose of the study. Please rewrite it. NO results of the study can be reported in this section.
Line 57: “In diabetic patients, skin ulceration of the foot is a stern medical condition which can lead to gangrene, infection or amputation.”
The authors must describe the different procedures reported in literature to avoid these complications, in particular plantar foot ulcers and Charcot foot, quoting:
· Minimally Invasive Distal Metatarsal Diaphyseal Osteotomy (DMDO) for Chronic Plantar Diabetic Foot Ulcers. Foot Ankle Int. 2018 Jan;39(1):83-92. doi: 10.1177/1071100717735640. Epub 2017 Nov 7.
· Mini Invasive Floating Metatarsal Osteotomy for Diabetic Foot Ulcers Under the First Metatarsal Head: A Case Series. The international journal of lower extremity wounds 2020, 10.1177/1534734620934579, 1534734620934579, doi:10.1177/1534734620934579.
· Minimally Invasive Surgery for Tibiotalocalcaneal Arthrodesis Using a Retrograde Intramedullary Nail: Preliminary Results of an Innovative Modified Technique. J Foot Ankle Surg. 2016 Nov-Dec;55(6):1130-1138. doi: 10.1053/j.jfas.2016.06.002. Epub 2016 Aug 11
· Minimally invasive metatarsal osteotomies (Mimos) for the treatment of plantar diabetic forefoot ulcers (pdfus): A systematic review and meta‐analysis with meta‐regressions. Applied Sciences (Switzerland). Open AccessVolume 11, Issue 20October-2 2021 Article number 9628. DOI 10.3390/app11209628.
Methods
This section does not contain enough information to understand but lack of some information to possibly repeat the study. Further, the manuscript does not reflect the Strobe Statement-Checklist for cohort studies. Please read these guidelines for articles before resubmitting the revision version.
Ethical statement must be reported. The article cannot be accepted without it and “not applicable” as reported in the manuscript statements is not the right reason to justify its absence. Please provide it!
Who did perform the statistical analysis? The same authors or an independent statistician? Please provide it.
Results
The results presented are quite complete, although they do not reflect the MM section as the Strobe guidelines were not respected.
Discussion
The length and content of the discussion communicates the main information of the paper but the results presented have not been discussed adequately with data provide in literature. Please rewrite it commenting more specifically the results you obtained with your study. You could add more references in this section.
Line 181: Diabetic foot lesions are susceptible to gangrene, ulceration and other complications.
Please, develop this part quoting:
· Distal Metatarsal Osteotomies for Chronic Plantar Diabetic Foot Ulcers. Foot Ankle Clin. 2022 Sep;27(3):545-566. doi: 10.1016/j.fcl.2022.02.003. Epub 2022 Aug 6.
· Laffenêtre, O.; Perera, A. Distal Minimally Invasive Metatarsal Osteotomy ("DMMO" Procedure). Foot and ankle clinics 2019, 24, 615-625, doi:10.1016/j.fcl.2019.08.011.
· Minimally Invasive Surgery: Osteotomies for Diabetic Foot Disease. Foot Ankle Clin. 2020 Sep;25(3):441-460. doi: 10.1016/j.fcl.2020.05.006. Epub 2020 Jul 9
The author provided the limitations of the study, you could also add future potential studies necessary to further clarify the matter.
Conclusions
There is not a proper conclusion section, please rewrite it.
References
The references are not up to date. Hence, if not strictly necessary, replace those before 2012 and add that suggested previously.
Tables and Figures
The number and quality of tables and figures are appropriate to transmit the main information of the paper.
Author Response
Dear reviewer,
Thank you for your observations.
I have changed the text size in the background section and I modified the keyword in alphabetic order.
The purpose of the study was to evaluate the data from patients admitted in our Surgery Clinic with type 2 diabetes complications necessitating surgery, through a retrospective study, as I mentioned in Material and Methods.
We performed only minor or major amputations because they came in emergency with diabetic foot infections. I am a general and vascular surgeon. My patients came to me after the osteotomies were performed, with complications, and the last option was incision, large debridment and amputation, those are the patient from this study.
Prophliaxys in my opinion should including as minimally invasive distal metatarsal diaphysis osteotomies and tibiotalocalcaneal arthrodesis using a retrograde intramedullary nail for chronic plantar DFU, and I mentioned it on the paper and I also cited as you recommended.
About the material and methods, I mentioned before de the conclusions that the study has some limitations, and as far as I am concerned I will try to continue this study with another one with a large number of patients.
Ethical statement (in romanian) is attached, obtained from the Hospital.
The statistical analysis was performed by one of the authors, Tamas Ilyes.
I also wrote about other Romanian studies regarding the costs and complications of DFU in the Discussions part. I wrote again the Conclusions. The references ar now up to date.

Reviewer 2 Report
This is an interesting retrospective study conducted on a cohort of patients affected by type 2 diabetes necessitating surgery for diabetic foot lesions; the characterization of the cohort permitted to obtain data and evaluate the role of several parameters, including age, complication rates, foot ulcer bacteriology, intensive care admission and urban or rural environment. The acquired knowledge permitted to identify most relevant risk factors suitable of prevention with simple interventions, which can lead to reduced need of radical treatments such as foot amputation. Data are clearly presented and discussed. Some points, hereafter suggested, could be considered to improve the merit of the work.
Line 92: The authors have used both OR and RR for describing their data. Some comment on the use of both risk estimates may be provided. According to specific literature (for instance: Ranganathan P, Aggarwal R, Pramesh CS. Common pitfalls in statistical analysis: Odds versus risk. Perspect. Clin. Res. 2015;6(4):222-4. doi: 10.4103/2229-3485.167092), RR cannot be used to infer about the population, when the number of exposed people is not available, as in retrospective studies; by contrast, when the outcome is not rare in the population, the OR will overstate the effect (Sedgwick P. Relative risks versus odds ratios. BMJ 2014;348:g1407) and may be not appropriate to estimate the relative risk of the disease.
Section 3.2: Which is the reason for distinguishing patients according to cut-off of 68 years of age? Also at lines 150, 154 an explanation could be provided while discussing the data.
Tables: the 95%CI for RR does not correspond to what is obtained by using any of the several calculators available also online; for example, with the tool on the Campbell Collaboration website (https://www.campbellcollaboration.org/escalc/html/EffectSizeCalculator-OR1.php), for the post-operative complications (Table 4, first item) the RR is 2.2 with 95%CI 0.8696-5.5655. Same values are obtained with at least 5 different online tools. Did the authors consider any correction factor to obtain their 95%CI interval? In such a case, the method should be reported.
Lines 119-120: There are no reported data showing the referred absence of differences considering gender (for instance, an additional supplementary material table could be helpful). Did the authors consider the fact that the low number of females could affect the power of the statistical analysis regarding gender-related differences? See also line 214 of discussion.
Line 142: The term 'diabetic' as a noun should be avoided. Preferred terminology is, for example, 'person with diabetes'. See for instance: Dickinson et al., The Use of Language in Diabetes Care and Education, Diabetes Care 2017;40:1790–1799 | https://doi.org/10.2337/dci17-0041 - Table 4.
Line 183: “out of that, 2.05% were people affected by DFUs”
Conclusions: perhaps a more definite sentence would be appropriate to finalize the paper. Just as a suggestion, I would transfer at the end of the paragraph the sentence now at lines 218-219, and include also here the sentence found at lines 27-28 of the abstract (“Identifying individuals at risk and their demographic and medical characteristics is vital for implementing prophylaxis programs and can significantly improve patients outcomes and quality of life, as well reduce economic burden for healthcare systems.”).
Line 230: Normally, approval from a review board (Institutional Review Board or Ethics Committee) is required for all studies involving people, medical record, and human samples, albeit the investigation is a retrospective study. If ethical approval is not required, the local or national legislation should be quoted.
Author Response
Dear reviewer,
Thank you for your observations.
The purpose of the study was to evaluate the data from patients admitted in our Surgery Clinic with type 2 diabetes complications necessitating surgery, through a retrospective study, as I mentioned in Material and Methods.
About the material and methods, I mentioned before the conclusions that the study has some limitations, and as far as I am concerned I will try to continue this study with another one with a large number of patients.
Ethical statement (in romanian) is attached, obtained from the Hospital.
The statistical analysis was performed by one of the authors, Tamas Ilyes.
I also wrote about other Romanian studies regarding the costs and complications of DFU in the Discussions part. I wrote again the Conclusions as you recommended. The references ar now up to date.

Reviewer 3 Report
A study of 69 patients is too small to have any clinical meaning. They have studied patients admitted for surgical procedures which limit the usefulness. Moreover they have excluded those who had adverse outcomes (death) during hospital admission which is significant reporting bias.
The aims of the study are not precisely presented. Is it a audit of patients presenting with foot complications?
Most of the cited literature related to foot morbidity and mortality pertains to US. The infrastructure and foot care facilities are disparate in different parts of the word. I believe that they might not be same in Romania as US. Kindly cite outcome studies from different geographical areas of the world.
No information is provided about who collected information, how patients were assessed, diabetic complications and co morbidities. Definitions used for foot complications are not provided. What was the criteria for ICU admission. What is the kind of ICU, is it post operative high dependency unit? Or something else.
What do they mean by bacteriology assay? How bacteriology was performed? What sample was used from what kind of patients? No information is provided about classification of DFU, or gangrene, its severity etc.
Figure 1 should be deleted.
Conclusions are nit sync with the aims and results. The authors say diabetic foot complications are costly but they have not presented any information or comparison of costs in the manuscript.
Author Response
Dear reviewer,
Thank you for your observations.
About the small number of patients, I mentioned before de the conclusions that the study has some limitations, and as far as I am concerned I will try to continue this study with another one with a large number of patients.
I am a general and vascular surgeon and the study was perform on a Surgery Department. We performed minor or major amputations because they came in our Clinic with diabetic foot infections.
Information regarding age, gender, living environment, duration of hospital stays, diagnosis, pre-operative bacteriological assay lesion exudate, post-operative complications, and whether these complications necessitated reintervention or ICU admission were pulled from the hospital data base. Records of patients of all ages with type 2 diabetes, admitted for either pre-scheduled or emergency surgery related to type 2 diabetes complications were included in this study.
The purpose of the study was to evaluate the data from patients admitted in our Surgery Clinic with type 2 diabetes complications necessitating surgery, through a retrospective study, as I mentioned in Material and Methods.
Ethical statement (in romanian) is attached, obtained from the Hospital.
The statistical analysis was performed by one of the authors, Tamas Ilyes.
I also presented the WIfI classification of The Society for Vascular Surgery for the threatened lower limb (2020).
The ICU is a postoperative intermediate care type.
The bacteriology was collected from the patient's ulcers with swabs and performed in the Hospital Lab.
I also wrote about other Romanian studies regarding the costs and morbidity of DFU in the Discussions part.
I deleted the picture. I wrote again the Conclusions.
The references ar now up to date.

Round 2
Reviewer 1 Report
The authors answered my comments properly.
Well done!
Author Response
Dear reviewer,
As another reviewer requested we performed some changes by adding some discussions and citations, and we rewrote the conclusions.
Thank you!

Reviewer 3 Report
The manuscript still needs much improvement
Abstract needs rewriting as it does not mention how many patients were studied, and main findings. The conclusions can be better presented instead of generalised statements like" s. Identifying risk factors which lead to foot complications in diabetic patients 27 and setting up prevention programs can significantly improve patients’ outcome, quality 28 of life and reduce the economic burden for healthcare systems" . None of this is studied in the present study so should be deleted.
As mentioned previously need to discuss other studies related to outcome of DFI like “J Foot Ankle Res 13, 16 (2020). https://doi.org/10.1186/s13047-020-00383-2”; "Diabetes Res Clin Pract. 2020 Apr;162:108113. doi: 10.1016/j.diabres.2020.108113" ; “J Diabetes Res. 2016;2016:2879809. doi:10.1155/2016/2879809”
The wifi need not to be discussed in introduction , instead authors need to provide WIFI grade for their patients. If not available then mention in limitations.
How many patients had charcot deformities? They may discuss literature regarding risk of amputations in charcot like " Sohn MW, et al. Lower-extremity amputation risk after charcot arthropathy and diabetic foot ulcer. Diabetes Care. 2010;33(1):98-100. doi:10.2337/dc09-1497"; van Baal J, et al. Mortality associated with acute Charcot foot and neuropathic foot ulceration. Diabetes Care. 2010;33(5):1086-1089. doi:10.2337/dc09-1428
Results: saying bacteriology positive and negative is not appropriate. The authors should provide details of bacterial isolates. Moreover, swab culture is not standard practice. They should have taken tissue or bone depending upon the involved tissue. What kind of bacteria are commonly isolated in their setting. the authors should discuss some pertinent literature related to bacterial isolates form other settings like " Atlaw A, Kebede HB, Abdela AA, Woldeamanuel Y. Bacterial isolates from diabetic foot ulcers and their antimicrobial resistance profile from selected hospitals in Addis Ababa, Ethiopia. Front Endocrinol (Lausanne). 2022;13:987487. Published 2022 Aug 31. doi:10.3389/fendo.2022.987487; J Diabetes Complications. 2017 Feb;31(2):407-412. doi: 10.1016/j.jdiacomp.2016.11.001" etc.
Please rephrase/rewrite conclusion as this is not what authors intended to study or they found.
Author Response
Dear reviewer,
Thank you for your observations.
With all due respect regarding that - Abstract needs rewriting as it does not mention how many patients were studied, and main findings - in the abstract we mentioned that: “In this retrospective study we analyzed and characterized a cohort of 69 patients and their diabetes related foot complications. The main characteristics of our cohort were as follows: older age at diagnosis (mean age 66); higher incidence of diabetes in males; with a dominant urban patient population. The most frequent complications of the lower extremity were ulcerations and gangrene. Moreover, in our study, 35% of patients required surgical reintervention and 27% suffered from complications while 13% required ICU admission”
As you suggested we discussed about the mortality in DFU - “J Foot Ankle Res 13, 16 (2020). https://doi.org/10.1186/s13047-020-00383-2” in the Discussion part and the limb amputation and mortality after the first neuropathic diabetic foot ulcer "Diabetes Res Clin Pract. 2020 Apr;162:108113. doi: 10.1016/j.diabres.2020.108113".
We moved the WIFI classification at the discussions and we mentioned in the limitations that we didn’t evaluated the WIFI grade of our patients.
In our study 10% of patients presented Charcot arthropaty. We discussed the literature regarding risk of amputations in charcot like " Sohn MW, et al. Lower-extremity amputation risk after charcot arthropathy and diabetic foot ulcer. Diabetes Care. 2010;33(1):98-100. doi:10.2337/dc09-1497".
Regarding the bacteriology the most commonly bacteria isolated in our patients was S. aureus and we discussed from the literature related to bacterial isolates form other settings like " Atlaw A, Kebede HB, Abdela AA, Woldeamanuel Y. Bacterial isolates from diabetic foot ulcers and their antimicrobial resistance profile from selected hospitals in Addis Ababa, Ethiopia. Front Endocrinol (Lausanne). 2022;13:987487.
We rewrote the conclusions as follows:
Early identification and the appropriate medical and surgical treatment of the complications associated with diabetic foot disease are important, because they still remain common, complex and costly. 35% of patients with diabetic foot complications (infections, ulcerations, gangrene) required surgical reintervention and 27% suffered from complications while 13% required ICU admission. The following factors have been found relevant for diabetic foot complications like older age (mean age 66), males and urban patient population.
